# Development of a UV Index Sensor-Based Portable Measurement Device with the EUVB Ratio of Natural Light

**DOI:** 10.3390/s19040754

**Published:** 2019-02-13

**Authors:** Dae-Hwan Park, Seung-Taek Oh, Jae-Hyun Lim

**Affiliations:** 1Department of Computer Science & Engineering, Kongju National Uiniversity, Cheonan-si, Chungcheongnam-do 32588, Korea; glow153@smail.kongju.ac.kr; 2Smart Natural Space Research Center, Kongju National University, Cheonan-si, Chungcheongnam-do 32588, Korea; ost73@kongju.ac.kr

**Keywords:** UV, portable measurement device, EUVB ratio, erythema weight, UV index sensor

## Abstract

Ultraviolet (UV) rays are electromagnetic waves that account for about 5% of solar light, and when overexposed, they pose malevolent effects on human skin and health. However, with recent reports on the beneficial effects of some wavelength bands of UV rays, people’s interest in UV information has increased. This has resulted in requiring not just simple information, such as the amount of UV or UV index (UVI), but detailed UV information that directly affects health, such as EUVB (erythemally weighted UVB). However, calculating EUVB, which can be done by applying the erythemal weighted function on the intensity value in wavelength, requires specialized optical measurement devices, which cannot be easily accessed by the general public; furthermore, public institutions’ UV information services do not offer EUVB information for individuals. Therefore, the present study proposes a UVI sensor-based portable measurement device, with which the general public can have easy access to UV-related information. The proposed device comprises a UVI sensor that can measure the intensity of erythemal UV radiation, a Bluetooth Low Energy (BLE) module that supports communication, and a micro controller unit (MCU) for key operations. In addition, it applies the ratio of EUVB by month/time, resulting from the actual analysis of natural light to calculate the EUVB and provides the amount of UVI and EUVB to check if they meet conditions required for outdoor activities through the device and smartphone applications. The applicability of the proposed device was verified by the measurement performance comparison test with the standard device, a spectrometer (CAS 140 CT), which showed an average error of 0.045 for UVI and 0.0014 W/m^2^. The proposed device’s offering of UV-related information such as UVI and EUVB to the user is expected to prevent potential damage due to exposure to UV and to support healthy outdoor activities.

## 1. Introduction

The solar radiation energy emitted from the sun and reaching the earth is delivered in electromagnetic waveforms, such as ultraviolet (UV) rays, visible rays, and infrared rays. Among them, UV rays, which account for about 5%, are divided into UVA (400–315 nm), UVB (315–280 nm), and UVC (280–100 nm) based on their wavelength [1,2]. While relatively shorter in wavelength and stronger in energy, UVC is completely absorbed into the ozone and other oxygen particles in the stratosphere; thus, it barely reaches the earth’s surface [3,4]. However, UVA and UVB do reach the surface and affect organisms considerably. UVA penetrates deeply into the dermal layer of the skin, promoting skin-aging and forming winkles [5,6]. Similar to UVC, most of UVB is absorbed into the ozone in the stratosphere, while only 10%–30% of UVB reaches the surface. However, its biological effects are known to be larger than that of UVA. UVB causes erythema, pigmentation, and optical aging, damages cell membranes and it can cause skin cancer in severe cases [7,8,9]. However, UVB is also known to play positive roles, including supporting the synthesis of vitamin D in the human body [10,11]. Such effects of UV rays on the human body are proportional to the erythemally weighted UV radiation (EUV), an integral value of the UV’s effect on erythema by wavelength.

As the direct effects of UV on human health have been unveiled and people’s interest has increased, various efforts have been made to offer UV-related information. The World Health Organization (WHO) has designed and proposed the UV index (UVI), which allows for an easy identification of the relative size of the amount of the UV radiation that reaches the earth’s surface. WHO advises to take measures to protect the skin by wearing long sleeved shirts or applying sun screen creams when the UVI is between 3 and 7, while refraining from doing any outdoor activities when it is over 7 [2]. Based on such advisory standards, many countries and organizations have offered UVI-based services. The Tropospheric Emission Monitoring Internet Service of the European Space Agency, located in the Netherlands, offers a real-time global UVI information service based on climatic data measured by satellites [12], and South Korea offers a service to monitor hourly UVI by region between March and October, in which the risk level of the UV increases significantly [13]. Moreover, UV information services use UV data offered by the government or agencies through which users can access UV information easily through web or smartphone applications [14,15]. However, the aforementioned UV information services using the wide regional UVI data offered by meteorological agencies cannot reflect environmental elements such as local climatic conditions at users’ locations, nor they can offer detailed information such as erythemally weighted UVB (EUVB) other than UVI as their main objective is to provide advice on outdoor activities to reduce negative effects when exposed to UV; furthermore, they do not contribute to the promotion of health through the appropriate level of exposure to natural light. Some researchers have studied the effect of the UV bands (280 to 380 mm) on human skin by measuring the intensity of natural light by wavelength with a special optical measuring instrument, i.e., a spectrometer [16]. While such special measuring equipment allows accurate calculation of the broadband UV radiation, it is difficult for general users to use such devices as they are expensive and require specialized knowledge to use. Moreover, such devices with poor mobility and portability due to their large size are made with sensitive materials, thus users must handle them with extra care, indicating that they are not easy to use in daily life. Some have proposed a potable spectrometer through which users can measure some UV and visible broadband using a small spectral sensor considering users’ convenience [17]. However, it would still require special knowledge to interpret any data measured.

Recently, portable small UV devices have been released in the market. An accessory using a UVI sensor or a band type portable UV-measuring device has been developed [18], and studies have been carried out to apply UVA and UVB sensors together to provide various types of UV-related information [19]. However, though these devices provided users with a UV index value, UVA or UVB values did not provide the EUVB information needed to calculate the amount of vitamin D in the body. Sundroid has provided the UVI through a smartphone application by measuring each radiation using the UVA sensor and UVB sensor and then applying the ratio of the EUVA and EUVB based on the pre-calculated erythemal weight [20]. However, this method does not consider the ratio of UVA and UVB changed by season and hour; thus, it is limited in offering accurate real-time based UV information. Another study has focused on a UVI and Particulate Matter alert service that uses both the user-measured values and the real-time UV information provided by meteorological agencies to enhance the accuracy of real-time information [15]. However, due to the limitations of the small UV sensors, this service could not offer EUV-related information to which the erythemal weight is applied. As such, existing UV information services have provided limited services on UVI values and corresponding outdoor activities without providing any EUVB information closely related to human health. Furthermore, these services do not offer real-time UV and EUVB information reflecting each user’s period of activity and regional characteristics.

Therefore, the present study proposes a UVI sensor-based portable device implemented with the natural light’s EUVB ratio to support users’ safe exposure to sunlight. The proposed device is designed in a portable device form for easy portability and it is linked with a smartphone application to inform users whether the daily advised amount of outdoor activity has been met. In order to achieve this, the portable UV measurement device comprises a UVI sensor measuring the total UVI, a Bluetooth Low Energy (BLE) module that supports wireless communication with the smartphone application, a micro controller unit (MCU, Arduino, Italy) for performing operations with the raw values acquired from the sensor, a display module offering the measured UV information visually, and a rechargeable battery module for supplying power. In addition, it is equipped with the monthly/hourly EUVB ratio index, which helps to calculate the EUVB through the UVI values in the MCU. For the improvement of the measurement performance of the device, it goes through the correction process using special measuring equipment, i.e., a spectrometer (CAS 140 CT, Instrument Systems, Munich, Germany). Later, the proposed device’s measurement performance and effectiveness are verified through the actual measurement test between the proposed device and the standard device. By providing the UVI and EUVB information surrounding the user, hopefully, the proposed device will support the users with healthy outdoor activities through preventing potential damage from their exposure to UV.

## 2. Methods: Monthly/Hourly EUVB Ratio of Natural Light 

The UV rays in solar light show different traits depending on location based on the latitudinal and longitudinal information, as well as regions such as urban or coastal regions. Moreover, even with the same UVI intensity values measured, the ratios of the UVA and UVB as well as the EUVA and EUVB, to which the erythemal weighting is applied, show different results based on month and time of the day. This study calculated the intensity of the EUVA and EUVB by applying the monthly and hourly ratio of the concerned region to the UVI intensity values acquired from the sensor. To create the EUVB ratio index by month and hour, this study measured the spectral luminosity of natural light by wavelength and analyzed the characteristics of UV radiation. The actual measurement of natural light was conducted at the rooftop of a 10-story building located at K University (latitude: 36.85°, longitude: 127.15°), Cheonan, Chungcheongnam-do, from April 2017 to March 2018, every day except on rainy days. The measurement of solar light was performed from sunrise to sunset every minute with a spectrometer (CAS 140 CT) using a tracking device, which moves the sensing element based on the altitude of the sun. From the daily spectral irradiance data measured, the ratio of the EUVB to the total EUV radiation was calculated to derive the daily EUVB data. The EUVA and the EUVB were derived by applying the erythemal weighting function to the intensity value ($E_{\lambda }$) by the wavelength of 315 to 400 nm, and 280 to 315 nm, respectively, as shown in Equation (1). Here the erythemal weighting function $\left(s_{er}\right)$ uses Equation (2), defined by in CIE209: 2014 [21].
(1)$$\begin{array}{c} EUVA=\overset{400}{\underset{315}{\int }}E_{\lambda }s_{er}\left(\lambda \right)\;d\lambda \\ EUVB=\overset{315}{\underset{280}{\int }}E_{\lambda }s_{er}\left(\lambda \right)\;d\lambda \end{array}$$
(2)$$s_{er}\left(\lambda \right)=\left\{\begin{array}{ll} 1 & \left(250\leq \lambda \leq 298\right)\\ 10^{0.094\left(298-\lambda \right)} & \left(298\leq \lambda \leq 328\right)\\ 10^{0.015\left(140-\lambda \right)} & \left(328\leq \lambda \leq 400\right) \end{array}\;\;\;(\lambda :\;\text{wavelength [nm]}\right)$$

To create the monthly and hourly ratio index, the daily data that had poor measurement data due to clouds, fog, aerosols, and partial omission were excluded, and the ratio of each daily EUVB was collected by month and hour. Figure 1 shows the collected monthly EUVA and EUVB intensity values.

The distribution of EUVA was between 0 and 0.06 W/m^2^, whereas that of EUVB was between 0 and 0.17 W/m^2^, showing that the degree of the effect of EUVB on the human body was up to 300% higher than that of EUVA. Moreover, the EUVB was found to be the lowest at 0.02 in winter and highest at 0.17 in summer by comparing the maximum values at noon. Furthermore, the values of EUVA and EUVB were shown to be different by month and hour, and the hour at which the EUVB ratio became the highest in a day was around 12:00 in all months. Although there was a slight difference by month, the data had a similar cycle of the daily pattern between 07:00 and 17:00, except in July when a slightly different pattern was observed due to the rainy season. The value 0 collected in some time zones attributes to the data being out of the effective time scope due to the sunrise and sunset in the month. Then, the shortest zone between sunrise and sunset during each month by considering the difference in daily sunrise and sunset times due to the revolution of the earth was selected as effective time scope for selecting the index. Later within the effective time scope of the monthly dataset, the arithmetic mean of the daily EUVB data was calculated and used for deriving the monthly/hourly EUVB ratio index, as shown in Figure 2. When the average EUVB ratio between 07:00 and 17:00 (the common effective time scope) was compared, the lowest ratio was observed for December at 0.49 and the highest at 0.66 for July. In addition, the comparison of the total hours in each month showed that the EUVB ratio was minimal at 0.16 at 05:00 in April and maximal at 0.74 at 12:00 in August, resulting in the largest difference in the monthly/hourly EUVB ratio up to 58%.

## 3. Methods: UVI Sensor-Based Portable Device 

The portable measurement device was developed to provide UV-related information through a smartphone application by calculating the amount of UVI and EUVB through the MCU after measuring the intensity of UV near users through the UVI sensor. Moreover, it was designed to check the amount of UVI and EUVB in real-time through its display module and provide additional UV-related services such as whether sufficient outdoor activities have been performed based on the cumulated amount of EUVB through a smartphone application. For this, the proposed device was equipped with the BLE module that supports wireless communication as well as a rechargeable battery module. Figure 3 shows the overview of the smartphone application that supports the portable measurement device and service. 

### 3.1. Hardware Design and Implementation 

The portable UV device used the TOCON E2 sensor to measure the UVI. It supports the UVI measurement to which action spectrum for erythema is applied. Moreover, it used Bluetooth module HM-10 for wireless communication to reduce power consumption as well as ATmega328-based Arduino Nano for the MCU, which offered identical performance to the controller used for more widely used Arduino Uno, but is as small as one third of the size. Thus, it was easy to manufacture in a portable device form. Additionally, the rechargeable battery module comprised the power supply module equipped with a step-up regulator and a 500 mAh lithium-ion polymer battery. Table 1 shows the hardware specifications of the device.

Figure 4 shows the circuit diagram of the UV portable device. The MCU was directly connected to a PC and firmware upgrade was performed via RS232C communication. Five V power was applied to the UVI sensor, whereas 3.3 V power was applied to the BLE module. The output value of the UV sensor was connected to the analog pin of the MCU and the RX, TX pins of the BLE module were connected to the digital pins of the MCU to send data. The display module received data with the MCU via SPI communication. Since the SPI communication requires Serial Clock (SCK), Master Out Slave In (MOSI), Reset (RST), Data/Command (DC), and Chip Select (CS), each of the signals was connected with each of the five digital terminals of the MCU. The + pole of the rechargeable battery module was connected to the Vin pin of the MCU to supply power to the MCU while the − pole was connected to the GND pin.

### 3.2. Firmware Design and Implementation 

The firmware implements the calculation and adjustment of the UVI based on the values measured through the sensor and the calculation of the EUVB based on the monthly/hourly EUVB ratio index. It also offers UV information via the display module as well as the communication support for connecting with a smartphone. Figure 5 shows the main process flow of the firmware.

The applied voltage for the UVI sensor used in the device is 5 V. Since it was connected to the analog pin of the MCU, the UVI sensor received voltage values between 0 and 5 V after they were converted to values from 0 to 1023. Since 1 UVI outputs 170 mV, the UVI sensor can measure up to 30 UVI [22]. Based on this, Equation (3) can be derived for converting the value acquired from the UVI sensor to UVI in the MCU. However, the measurement scope of the UVI sensor used in the proposed device is 297–391 nm, which is different from 280–400 nm, the standard UVI measurement scope of WHO. To derive an equation that corrects the error caused by such a difference, UV was measured under the identical environment as that of the standard device, then the measured results needed to be compared and analyzed. Therefore, this study derives a correction equation, Equation (4), based on the data collected through actually measuring the natural light with the UVI sensor and a spectrometer (CAS 140 CT) under the identical environment and then applying it by implementing through the firmware.
(3)$$UVI=5000\times \frac{a}{1023}\times \frac{1}{170}$$
(4)$$UVI=1.0097\times \left(5000\times \frac{a}{1023}\times \frac{1}{170}\right)+0.0029\;\;\;(a:\;\text{input value from analog pin})$$

WHO proposes a method of calculating the UVI by multiplying constant value k to EUV, which is shown in Equation (3) [2]. Here, EUV is the total erythemal UV radiation reaching the surface, which is the same as the sum of EUVA and EUVB, and thus, EUVB can be derived from the multiplication of UVI and an optimum ratio. To calculate the amount of EUVB from the UVI, this study used the monthly-diurnal EUVB scale factor, which analyzed and collated the natural light’s UV radiation characteristics, to derive Equation (4), an EUVB calculation equation.
(5)$$UVI=k\cdot EUV\;\;\;\;\;\left[W/m^{2}\right]$$
(6)$$EUVB=UVI\cdot \frac{1}{k}\cdot MESF\;(k=40)$$

The resulting UVI and EUVB are offered to the user via the display module, and for more detailed UV-related services, the information is delivered to a smartphone via the BLE module. Figure 6 shows the communication protocol between the smartphone application and the proposed device.

The communication was divided into the reception protocol and the transmission protocol, each of which comprised 14 bytes. The 1 byte at the beginning and the end of the packet comprised STX (0x02) and ETX (0x03), respectively, and each packet contained the device serial number (DEV_ID) and checksum (CHKSUM). The transmission protocol packet included the 2 bytes of the UVI value measured through the UV sensor and 4 bytes of the EUVB value calculated by the monthly/hourly EUVB ratio. The upper 1 byte of the UVI data was the integer part of the UVI value that had been converted and corrected, while the lower 1 byte was the floating value. The reception protocol packet included 8 bytes of the date and time for the synchronization of the time on the portable UV device via the smartphone. The checksum, which is a value used to check the effectiveness of the transmission and reception packet, was derived by performing XOR calculation by 1 byte starting from the 3rd byte to 11th byte of the packet.

### 3.3. UV Information Service 

The UV information service comprises the device service provided through the display module and the detailed EUVB service offered via the smartphone application. The device service offered real-time UVI value and EUVB results to the user via the display module. The detailed EUVB service offered via the smartphone application included information such as the amount of real-time EUVB, hourly EUVB monitoring, estimated amount of synthesized vitamin D by user, and satisfaction of the outdoor activities’ duration based on the amount of the synthesized vitamin D. To calculate the amount of the synthesized vitamin D, Equation (7), which reflects the age, skin type, area of the body exposure, and other personal information of the user, was used [23]. Additionally, based on the estimated amount of the synthesized vitamin D, the advised duration of outdoor activities was calculated and offered.
(7)$$VitD=4000\cdot \frac{EUVB\cdot t}{MEDF\cdot SED}\cdot EA\cdot A\;\;\;\;\;\;\;\;\left[\mathrm{IU}\right]$$

EUVB is the EUVB value converted from UVI measured from the sensor and changes with time. In Equation (7), *t* is the duration of exposure (s), and SED is the standard erythemal dose, 100 J/m^2^. The required factors for calculating vitamin D using EUVB are minimum erythemal dose factor (MEDF), exposure area (EA), and age (A). MEDF used the values based on Fitzpatrick’s skin type [24]. The exposure area EA was calculated from the body surface area ratio of the Lund and Browder chart [25]. In addition, the age variable A was applied to the difference in vitamin D synthesis capability according to age [26]. Based on Equations (7) and (8) was derived to obtain the residual exposure time t necessary to meet the daily vitamin D recommended dose based on the current EUVB intensity offered.
(8)$$t=\frac{DVD\cdot MEDF\cdot SED}{4000\cdot EUVB\cdot EA\cdot A}\;\;\;\;\;\;\;\;\left[s\right]$$

In Equation (8), *DVD* means the amount of vitamin D that is deficit minus the amount of anticipated vitamin D synthesis till the present from the recommended daily amount of vitamin D. Information on vitamin D’s anticipated amount of synthesis and exposure time calculated from Equations (7) and (8), as well as UV-related information, were provided through the smartphone application. The smartphone application was set to allow users to set age, skin type, and current skin exposure area. For the exposed area, 0.15 was applied in spring and autumn, 0.35 in summer, and 0.1 in winter, considering seasonal clothing [23,24]. Figure 7 shows an example of the execution of the EUVB detailed service through the smartphone application. In this case, MEDF was set to 400 IU and age variable A was set to 1 assuming the user information was Asian, 20 years old. The EA of 0.15 was applied by assuming the exposed areas of face (3.5%), neck (1%), hand (3%), and arms (6%), respectively.

## 4. Results: Correction and Evaluation 

### 4.1. Correction Equation Creation Test 

The study conducted a test to correct the measurement errors caused by the difference between the measurement scope (297–391 nm) of the UVI sensor (TOCON E2) used in the device and the standard UVI calculation scope (280–400 nm). The test was conducted at the rooftop of a 10-story building in Kongju University, located in Cheonan, Chungcheongnam-do, which had an identical environment as the actual measurement environment for extracting the EUVB ratio index, on February 8, 2018, for 12 h from sunrise (07:27) to sunset (18:30). The spectral luminosity was measured with a spectrometer (CAS 140 CT), whereas the UVI value was measured every 30 min using the portable UV device. Later, based on the measurement data of the spectrometer, EUV was calculated using Equation (1) and the portable UV device was used to calculate the EUV as well. Table 2 shows the results of the daily EUV measurement tests that were calculated and measured by each device.

The test results showed that the absolute mean error of EUV calculated by the spectrometer and the EUV measured by the device was 0.0034 W/m^2^. To reduce the measurement error of the portable UV device using these values measured, a linear regression analysis was used to derive Equation (9).
(9)y=1.0097x+0.0029

Later, Equation (9) was used to correct the EUV. Figure 8 shows the EUV graphs before and after the use of the correction equation. The correction result showed 0.0017 W/m2 of mean error, indicating that the correction equation could reduce the error by over 50%.

### 4.2. EUVB Measurement Performance Evaluation 

In order to evaluate the measurement performance of EUVB, the measured results of EUVB and the portable UV device based on the measured results of the spectrometer (CAS 140 CT) were compared. In order to evaluate the measurement performance, natural light was measured after selecting each day of spring with relatively high ultraviolet intensity, and winter with relatively low ultraviolet intensity. Figure 9 shows the results of EUVB using the measured results of each seasonal UVI and the EUVB ratio indicator. The UVI measurement result showed an average difference of 0.045 compared to the reference instrument (CAS 140CT), showing that it coincided until the second digit below the decimal point. The EUVB calculated by the proposed method showed errors of 0.0010 and 0.0018 W/m^2^ in spring and winter, respectively. The average error of EUVB is 0.0014W/m^2^, which corresponds to 1% when compared with the maximum EUVB size (0.17W/m^2^) in the immediately preceding year within the region. Therefore, it is confirmed that accurate EUVB information can be provided and put to practical use with the proposed device.

## 5. Conclusion and Future Research

This study proposed a UVI sensor-based portable device based on the EUVB ratio of natural light. The hardware of the proposed device comprised a UVI sensor, BLE module, display module, MCU, and battery module. The firmware realized the UV sensing value-based UVI calculation and UVI correction function and calculation of the EUVB by applying EUVB ratio of natural light. To calculate the EUVB, the study measured and collated natural light’s UV data for one year from April 2017 to March 2018, which were analyzed to derive the EUVB ratio index by month and hour. The derived EUVB ratio index was applied to the calculation of the EUVB to offer real-time EUVB-related information services to the user. The UV information services offered intensity of the UVI and EUVB via the device as well as EUVB exposure monitoring, the amount of the synthesized vitamin D, and the outdoor activities guide for satisfying the suggested amount of the synthesized vitamin D through a smartphone application. To reduce the error caused by the difference between the measurement scope of the UVI sensor and that of the EUV regulated by WHO and CIE, the study measured the natural light for one day using the standard device (CAS 140 CT) and the UVI sensor-based portable device and derived a regression equation based on the actually measured data, which corrects the measurement error between the two devices. The derived regression equation was applied to the UVI calculation to improve the device’s measurement performance. In addition, the evaluation of the EUVB measurement performance verified the validity of the proposed services by showing the mean error of 0.0014 W/m^2^ compared to the results from the standard device (CAS 140 CT).

The proposed device’s offering of UV-related information such as UVI and EUVB to users is expected to prevent potential damage due to the exposure to UV and to support healthy outdoor activities. In future studies, in order to secure the portability of the proposed device, research for collecting the EUVB ratio data by position according to latitude and longitude is required, and research for the commercialization of the UVI sensor-based portable device using collected EUVB data is needed. In addition, further research is required on the development of healthy lighting systems that would consider the amount of individuals’ outdoor activity by linking the portable UV device to indoor lighting.

## Figures and Tables

**Figure 1 sensors-19-00754-f001:**
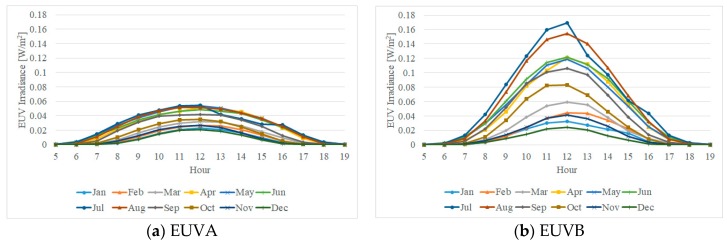
Current condition of the monthly erythemally weighted UV radiation (EUV) daily cycle (2017–2018; Cheonan, Chungcheongnam-do).

**Figure 2 sensors-19-00754-f002:**
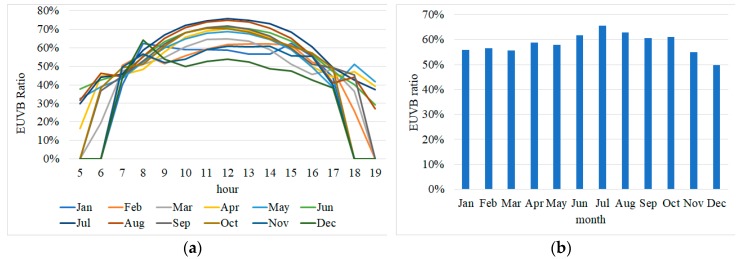
Monthly/hourly erythemally weighted UVB (EUVB) ratio index graph (**a**) and the average (**b**).

**Figure 3 sensors-19-00754-f003:**
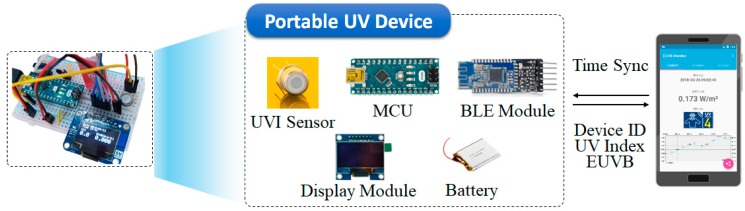
Portable measurement device and smartphone application. UVI: UV index; MCU: micro controller unit; BLE: Bluetooth Low Energy.

**Figure 4 sensors-19-00754-f004:**
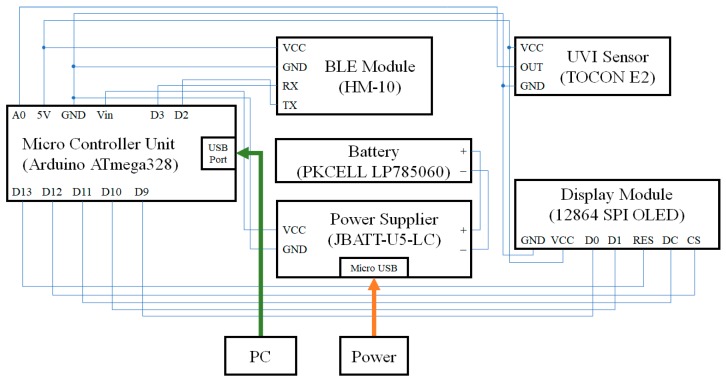
Circuit diagram of the portable measurement device. DC: Data/Command; CS: Chip Select.

**Figure 5 sensors-19-00754-f005:**
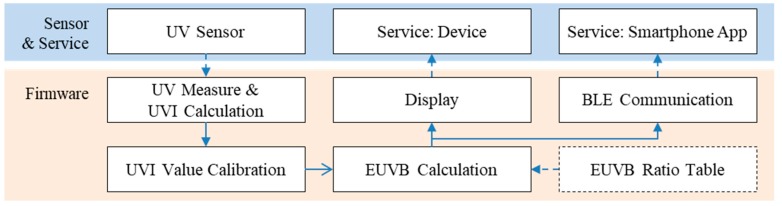
The main process flow of the firmware.

**Figure 6 sensors-19-00754-f006:**
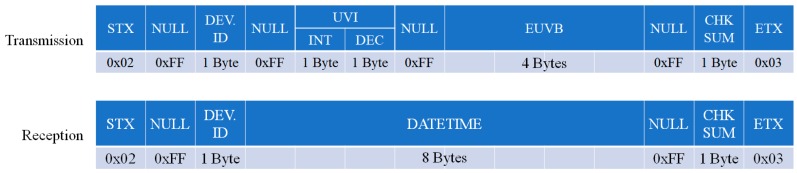
Transmission and reception protocol between the smartphone application and the device. DEV. ID: device serial number; CHK SUM: checksum.

**Figure 7 sensors-19-00754-f007:**
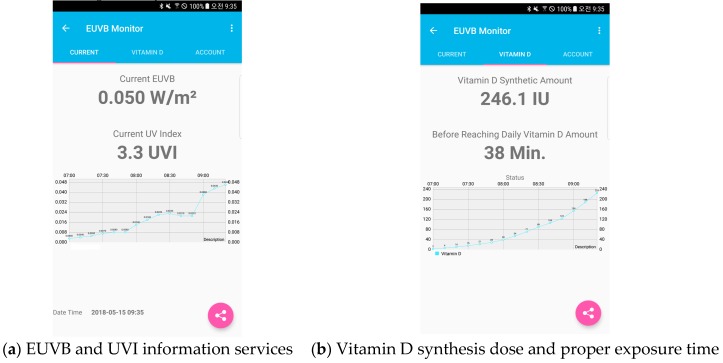
Smartphone application service UI (User Interface).

**Figure 8 sensors-19-00754-f008:**
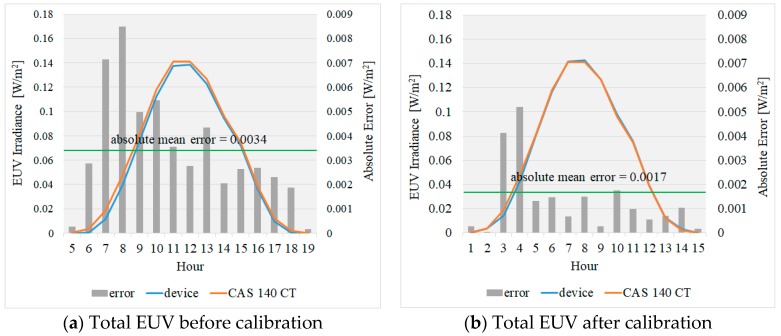
EUV graphs before and after the application of the compensation equation.

**Figure 9 sensors-19-00754-f009:**
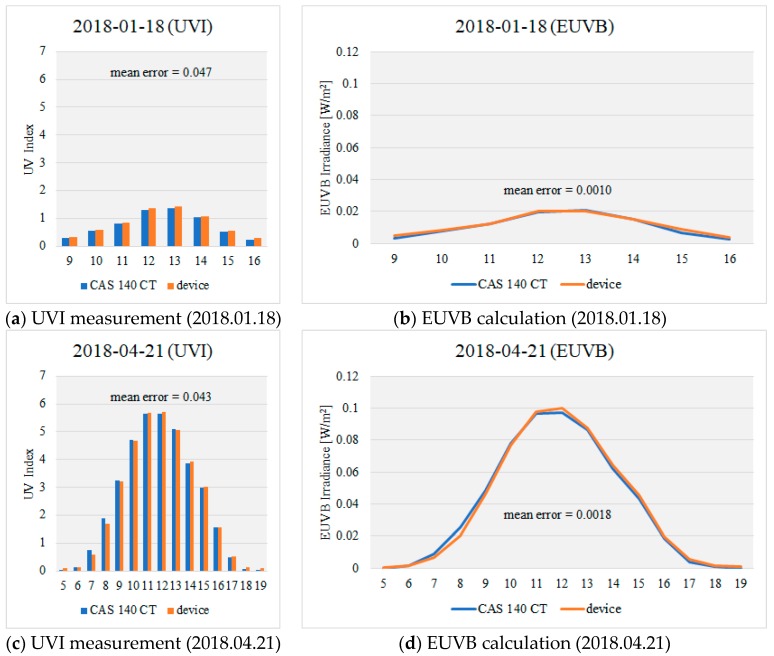
Graphs of the measurement results**.**

**Table 1 sensors-19-00754-t001:** Hardware specifications**.**

Type	Model Name	Details
UVI sensor	TOCON E2	Sensing factor: UV Index, Responsibility Range: 297~391 nm UV Index Measurement Range: 0~30 1 UVI results an output voltage 170 mV Operating Temperature: −25~85 °C
Microcontroller Unit	Arduino Nano	ATmega328
Display Module	12864 OLED LCD	Pixel: 128 × 64, Communication Protocol: SPI
BLE Module	HM-10	Communication Protocol: BLE (Bluetooth Low Energy)
Rechargeable Battery Module	PKCELL LP785060	Capacity: 500 mAh, Step-up Regulator

**Table 2 sensors-19-00754-t002:** Daily EUV measurement data (unit: W/m^2^).

Hour	CAS 140CT	Device	Abs. Error	Hour	CAS 140CT	Device	Abs. Error
5	0.000265	3.35E-07	0.000265	13	0.127031	0.122679	0.004352
6	0.003485	0.000621	0.002864	14	0.096203	0.094149	0.002054
7	0.018658	0.011525	0.007133	15	0.074512	0.071889	0.002623
8	0.047482	0.038996	0.008486	16	0.038775	0.036085	0.00269
9	0.08145	0.076481	0.004969	17	0.012062	0.009759	0.002303
10	0.117902	0.112449	0.005453	18	0.002144	0.000277	0.001867
11	0.14112	0.137568	0.003551	19	0.000163	0	0.000163
12	0.141011	0.138258	0.002753	Mean Absolute Error			**0.003435**

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
