# Peer review of "Development of a UV Index Sensor-Based Portable Measurement Device with the EUVB Ratio of Natural Light"

_sensors, 2019, doi:10.3390/s19040754_

Round 1

Reviewer 1 Report

The authors present a manuscript about a new device to measure ultra violet radiation for human skin protection. The paper is well written, and easy to follow. The system is well described, and good amount of background data are given.

The only point I wanted more attention is that the new system is still apparentely a single band sensor (297-391 nm), thus covering both UVA and UVB, and then corrected empirically to give the desired UVI and EUVB values. This maybe well enough for everyday purposes, but then it is not clear, how sensitive this is to extreme conditions (mountains, polar regions, strongly reflecting land surfaces, artificial illuminations etc.), or even normally varying atmosphere.

I see this paper valuable and recommend to publish.

Author Response

Dear Reviewer 1.

 I sincerely thank the reviewer who reviewed my paper.

Sincerely,
SeungTaek Oh.

Reviewer 2 Report

The authors present a portable device for the UV index measurement. The proposed architecture includes a portable measurement device & Smartphone Application. They have tested their device against actual readings.

The main idea is interesting; however, this work presents limited novelty since similar devices have been presented in the literature. Also, the testing of the device is rather limited.

Comments:

1) The authors do not present similar devices presented in the literature. Some examples:

https://www.semanticscholar.org/paper/A-Wireless-Embedded-Device-for-Personalized-Amini-Matthews/1340a783e68ce2793f062ce5d4b8f7d2256e9177

https://www.mdpi.com/2411-5134/3/2/26

The authors must include all related research attempts in the introduction of their manuscript.

2) However, the most important issue is to include a detailed qualitative comparison with similar approaches proposed in the literature in order to demonstrate the scientific added value of their work.

3) The authors have tested their device using data from a single day. The testing is hardly sufficient and should be expanded to cover a larger period of time.

4) The authors include personal information in their calculations, however it seems that the testing has been made using a single user. The authors must include additional users testing the device.

5) Also, the proper exposure time is calculated, however the authors do not describe how this is done. The authors must include a detailed description of this measurement (i.e. how this 11 Min. has occurred in fig.7b?). Also, how this 11 Min. is validated?

6) The authors should comment on the portability of their device.

Author Response

Dear Reviewer 2.

I sincerely thank the reviewer who reviewed my paper.

The answers and corrections to your points are arranged as follows.

1) The author does not suggest similar devices as presented in the literature. Some examples:

https://www.semanticscholar.org/paper/A-Wireless-Embedded-Device-for-Personalized-Amini-Matthews/1340a783e68ce2793f062ce5d4b8f7d2256e9177

https://www.mdpi.com/2411-5134/3/2/26

Authors should include all relevant research attempts when submitting manuscripts.

[Answer] Reference(18, 19) was added, and the related sentence was added and modified in the Introduction.

[Amended]

Line 77 → An accessory using UVI sensor or a band type portable UV-measuring device has been developed, and studies have been carried out to apply UVA and UVB sensor together to provide various types of UV related information. However, though these devices provided users with a UV index value, UVA or UVB values did not provide the EUVB information needed to calculate the amount of vitamin D in the body. 

2) However, the most important question is to include a detailed qualitative comparison with a similar approach proposed in the literature to demonstrate the scientific value-add of the study.

[Answer] Figure 9 shows the measurement results of UVI which used to be provided in the existing methods by adding the picture. The measurements results for comparison with the spectrometer (CAS 140 CT), which is also used as a reference device, are also expressed.

3) The author used the data to test the device in one day. Testing is almost insufficient and should be extended to cover longer periods.

[Answer] Test results of the proposed device was added. For the study on the commercialization of the proposed device in the future, calibration for all the seasons and long-term experiment of the device would be carried out.  

[Amended]

Line 295 → In order to evaluate the measurement performance of EUVB, the measured results of EUVB and portable UV device based on the measured results of spectroscopy (CAS 140 CT) were compared. In order to evaluate the measurement performance, natural light was measured after selecting each day of spring with relatively high ultraviolet intensity, and winter with relatively low ultraviolet intensity. Figure 9 shows the results of EUVB using the measured results of each seasonal UVI and the EUVB ratio indicator. The UVI measurement result showed an average difference of 0.045 compared to the reference instrument (CAS 140CT), showing that it coincided until the second digit below the decimal point. The EUVB calculated by the proposed method showed errors of 0.0010 and 0.0018 W/m2 in spring and winter, respectively. The average error of EUVB is 0.0014W/m2, which corresponds to 1% when compared with the maximum EUVB size (0.17W/m2) in the immediately preceding year within the region. Therefore, it is confirmed that accurate EUVB information can be provided and put to practical use with the proposed device. 

4) The authors include personal information in the calculations, but the tests appear to have been made using a single user. Authors should include additional users to test the device.

[Answer] The sentence expresses an example of a UV information service. In Section 3.3, a description of the categorization of the users according to skin color, amount of vitamin D synthesis according to skin exposure area and age, and references are added and amended to provide a calculation method for other users.  

5) Also, the appropriate exposure time is calculated, but the author does not explain how this is done. The author should include a detailed description of this measurement (eg how 11 minutes occurred in Figure 11b). Also, how about 11 minutes. Checked?

[Answer] The explanatory note related to Equation (8) was added to show how to calculate the proper exposure time. The UI display was also updated in Figure 7.

[Amended]

Line 250 → EUVB is the EUVB value converted from UVI measured from the sensor and changes with time.

Line 254 → The exposure area EA was calculated from the body surface area ratio of the Lund and Browder chart [25]. In addition, the age variable A was applied to the difference in vitamin D synthesis capability according to age [26]. Based on Equation (7), Equation (8) was derived to obtain the residual exposure time t necessary to meet the daily vitamin D recommended dose based on the current EUVB intensity. 

In Equation (8), DVD means the amount of vitamin D that is deficit minus the amount of anticipated vitamin D synthesis till the present from the recommended daily amount of vitamin D. Information on vitamin D’s anticipated amount of synthesis and exposure time calculated from Equations (7) and (8), as well as UV-related information, were provided through smartphone application. Smartphone application was set to allow users to set age, skin type, and current skin exposure area. For the exposed area, 0.15 was applied in spring and autumn, 0.35 in summer, and 0.1 in winter, considering seasonal clothing [23, 24]. Figure 7 shows an example of the execution of the EUVB detailed service through the smartphone application. In this case, MEDF was set to 400IU and age variable A was set to 1 assuming the user information was Asian, 20 years old. The EA of 0.15 was applied by assuming the exposed areas of face (3.5%), neck (1%), hand (3%), and arms (6%), respectively.  

6) The author should mention the portability of the device.

[Answer] Table 1 summarizes the UVI sensor specifications (including environmental conditions for measurements). In addition, the proposed device can provide information in any area when the EUVB ratio value according to latitude and longitude is applied.

[Amended]

Line 335 → In future studies, in order to secure the portability of the proposed device, research for collecting the EUVB ratio data by position according to latitude and longitude is required, and research for the commercialization of the UVI sensor-based portable device using collected EUVB data is needed.

Sincerely,
SeungTaek Oh.

Reviewer 3 Report

This work proposes a UVI sensor-based portable device implemented with the natural light's EUVB ratio to support users' safe exposure to sunlight, which includes a UVI sensor, a BLE module, a micro controller unit (MCU), a display module and a rechargeable battery module. The proposed device is designed in a portable device form for easy portability and it is linked with a smartphone application to inform users whether the daily advised amount of outdoor activity has been met. The results are attractive in fields of biological and medical applications and appear sounds. However, there are some specific points needed to be addressed. I recommend this manuscript a minor revision before the acceptance for publication.

Some comments:

This paper did not mention the stability of the portable device. Whether the long continuous operation will influence the accuracy of the device or not? The author should provide this result.

All the results of the portable device are obtained at the rooftop of a 10-story building located at K University, Cheonan, Chungcheongnam-do. The author should explain how to make this device applicable to other place in different climates.

The description of the UVI sensor seems to be too little. Additions of more details (eg. type, specification, property) are needed.

As saying in 3.3. UV information service, ‘Specifically, EA was 0.15 in spring and fall, 0.35 in summer, and 0.1 in winter.’ Author should explain the reason that why choose these value as ‘EA’.

Author Response

Dear Reviewer 3.

I sincerely thank the reviewer who reviewed my paper.

The answers and corrections to your points are arranged as follows.

1) This paper did not mention the stability of the portable device. Whether the long continuous operation will influence the accuracy of the device or not? The author should provide this result.

[Answer] The purpose of this paper is to present EUVB information, providing method for the calculation of vitamin D synthesis amount through UVI sensor-based device. Long-term use experiments will be conducted in the future for commercialization.

2) All the results of the portable device are obtained at the rooftop of a 10-story building located at K University, Cheonan, Chungcheongnam-do. The author should explain how to make this device applicable to other place in different climates.

[Answer] A description of the measurable environment of the sensing device was added.  

[Amended]

Line 335 → In future studies, in order to secure the portability of the proposed device, research for collecting the EUVB ratio data by position according to latitude and longitude is required, and research for the commercialization of the UVI sensor-based portable device using collected EUVB data is needed.

3) The description of the UVI sensor seems to be too little. Additions of more details (eg. type, specification, property) are needed.

[Answer] A detailed description of the UVI sensor was added in the Table 1.

4) As saying in 3.3. UV information service, ‘Specifically, EA was 0.15 in spring and fall, 0.35 in summer, and 0.1 in winter.’ Author should explain the reason that why choose these value as ‘EA’.

[Answer] This is an example of a UV information service. A calculation method for other users was provided by adding the categorization of the users depending on the skin color, the difference in the amount of vitamin D synthesis according to the exposed area of the skin and the age, and the related reference.

[Amended]

Line 254 → The exposure area EA was calculated from the body surface area ratio of the Lund and Browder chart [25]. In addition, the age variable A was applied to the difference in vitamin D synthesis capability according to age [26]. Based on Equation (7), Equation (8) was derived to obtain the residual exposure time t necessary to meet the daily vitamin D recommended dose based on the current EUVB intensity.  

In Equation (8), DVD means the amount of vitamin D that is deficit minus the amount of anticipated vitamin D synthesis till the present from the recommended daily amount of vitamin D. Information on vitamin D’s anticipated amount of synthesis and exposure time calculated from Equations (7) and (8), as well as UV-related information, were provided through smartphone application. Smartphone application was set to allow users to set age, skin type, and current skin exposure area. For the exposed area, 0.15 was applied in spring and autumn, 0.35 in summer, and 0.1 in winter, considering seasonal clothing [23, 24]. Figure 7 shows an example of the execution of the EUVB detailed service through the smartphone application. In this case, MEDF was set to 400IU and age variable A was set to 1 assuming the user information was Asian, 20 years old. The EA of 0.15 was applied by assuming the exposed areas of face (3.5%), neck (1%), hand (3%), and arms (6%), respectively.  

Sincerely,
SeungTaek Oh.

Round 2

Reviewer 2 Report

The provided answers are not for any of my comments!